# Altered Clock Gene Expression in Female APP/PS1 Mice and Aquaporin-Dependent Amyloid Accumulation in the Retina

**DOI:** 10.3390/ijms242115679

**Published:** 2023-10-27

**Authors:** Laura Carrero, Desireé Antequera, Ignacio Alcalde, Diego Megias, Lara Ordoñez-Gutierrez, Cristina Gutierrez, Jesús Merayo-Lloves, Francisco Wandosell, Cristina Municio, Eva Carro

**Affiliations:** 1Neurobiology of Alzheimer’s Disease Unit, Functional Unit for Research into Chronic Diseases, Instituto de Salud Carlos III, Network Centre for Biomedical Research in Neurodegenerative Diseases (CIBERNED), ISCIII, 28029 Madrid, Spain; carrero.olivas@gmail.com (L.C.); eeara@yahoo.es (D.A.); cristinaagutierrezdz@gmail.com (C.G.); 2PhD Program in Neuroscience, Autonoma de Madrid University, 28049 Madrid, Spain; 3Instituto Universitario Fernández-Vega, Universidad de Oviedo, Fundación de Investigación Oftalmológica, 28012 Oviedo, Spain; nacho.alcalde@fio.as (I.A.); merayo@fio.as (J.M.-L.); 4Instituto de Investigación Sanitaria del Principado de Asturias (ISPA), 33011 Oviedo, Spain; 5Advanced Optical Microscopy Unit, Unidades Centrales Científico-Técnicas, Instituto de Salud Carlos III, 28222 Madrid, Spain; diego.megias@isciii.es; 6Centro de Biología Molecular “Severo Ochoa” (CSIC-UAM), Universidad Autónoma de Madrid, Network Centre for Biomedical Research in Neurodegenerative Diseases (CIBERNED), ISCIII, 28029 Madrid, Spain; lordoniez@cbm.csic.es (L.O.-G.); fwandosell@cbm.csic.es (F.W.)

**Keywords:** Alzheimer’s disease, circadian rhythm, clock genes, retina, transgenic mice, hypothalamus, cerebral cortex, hippocampus, amyloid

## Abstract

Alzheimer’s disease (AD), the most prevalent form of dementia, is a neurodegenerative disorder characterized by different pathological symptomatology, including disrupted circadian rhythm. The regulation of circadian rhythm depends on the light information that is projected from the retina to the suprachiasmatic nucleus in the hypothalamus. Studies of AD patients and AD transgenic mice have revealed AD retinal pathology, including amyloid-β (Aβ) accumulation that can directly interfere with the regulation of the circadian cycle. Although the cause of AD pathology is poorly understood, one of the main risk factors for AD is female gender. Here, we found that female APP/PS1 mice at 6- and 12-months old display severe circadian rhythm disturbances and retinal pathological hallmarks, including Aβ deposits in retinal layers. Since brain Aβ transport is facilitated by aquaporin (AQP)4, the expression of AQPs were also explored in APP/PS1 retina to investigate a potential correlation between retinal Aβ deposits and AQPs expression. Important reductions in AQP1, AQP4, and AQP5 were detected in the retinal tissue of these transgenic mice, mainly at 6-months of age. Taken together, our findings suggest that abnormal transport of Aβ, mediated by impaired AQPs expression, contributes to the retinal degeneration in the early stages of AD.

## 1. Introduction

Alzheimer Disease (AD), the most common form of dementia, is a progressive and multifactorial disorder leading to progressive memory loss, cognitive deficits, and behavioral changes [1,2,3,4]. The main pathological and diagnostic features are the accumulation of two proteins: amyloid-β (Aβ) peptide, which aggregates into extracellular plaques, and hyper-phosphorylated tau, which forms intracellular neurofibrillary tangles [5]. Along with advanced age and apolipoprotein E (*APOE*)-4 genotype, epidemiologic and clinical studies have shown that female gender is a major risk factor for developing AD [6,7]. However, gender differences in the development and neuropathology of AD have received little attention, and the cellular and molecular mechanisms underlying these differences remain unknown.

In addition to cognitive decline, AD patients show circadian dysfunction [8,9]. In particular, circadian rhythm disturbances are associated with the functional disruption of biological clocks in the suprachiasmatic nucleus (SCN), the master circadian pacemaker [10], in the cerebral cortex rhythms [11], and in the hippocampus [12]. Three components are involved in the SCN’s ability to function as a brain clock, including input from the environment via a direct retinohypothalamic tract (RHT), an oscillating clock in the SCN, and output pathways to various target areas in nearby hypothalamic regions.

Circadian rhythm is associated with sex differences [13]. It has been reported that the timing of the expression levels of three core clock genes, *Per2*, *Per3*, and *Arnt1,* occur significantly earlier in women than in men [14]. Sex differences in SCN afferent pathways have been also described, including higher expression of estrogen receptors in the ventromedial nucleus of the hypothalamus [15].

However, there is a paucity of research on the sex differences associated with AD-related changes in the circadian rhythms and pathogenic mechanisms. For example, it has been shown that 2-month-old APP/PS1 mice exhibit altered expression of the clock genes *Per1*, *Per2*, *Cry1,* and *Cry2* compared with wild-type control mice in the medulla/pons [16]. Sleep duration was also reduced in female APP/PS1 mice, suggesting sleep fragmentation and reduced amplitude [17].

Sex differences in the pathological characteristics of transgenic AD mice have been demonstrated in numerous studies. For instance, immunohistochemical analysis revealed that female APP/PS1 mice carry a higher Aβ burden compared with male APP/PS1 mice [18,19,20]. Compared with male APP/PS1 mice, the females exhibited more severe cerebral amyloid angiopathy [21]. In female 3xTg-AD mice the accumulation of Aβ in brain tissue is reported to occur earlier and progress at an accelerated rate compared to male mice [22,23,24]. Since Aβ-peptide originates from a larger precursor, the amyloid precursor protein (APP) ubiquitously expressed, Aβ accumulation can be found in peripheral tissues, such as retina. Neuropathological hallmarks of AD, including Aβ accumulation, are shown in the retina of AD patients [25,26,27,28] and play a critical role in sending sensory information to the SCN and the subsequent regulation of the circadian rhythm. Recent studies suggest that the brain and retina not only share many structural, cellular, molecular, and functional similarities, but also follow a similar trajectory during AD progression [29]. Recent findings have identified an ocular glymphatic clearance route for Aβ clearance via a pathway dependent on the glial water channel aquaporin-4 (AQP4) [30]. Reduced AQP4 expression was described in the cerebral cortex of AD patients and was associated with increased Aβ accumulation [31]. Although the relevance of AQP4 has been reported in AD brains [32,33,34], expression changes in AQP4 and other AQPs expressed in the AD retina are unknown. 

Taken together, it is still necessary to further clarify the severity of circadian dysfunction and retinal amyloidosis in a female-experimental AD model, such as APP/PS1 mice. Our current study, together with our previous results on retinal neurodegeneration in male APP/PS1 mice [35], suggests that retinal amyloidosis in both females and males may be closely associated with decreased AQP expression, possibly leading to Aβ-induced pathological processes not only in the retina but also in the SCN. 

## 2. Results

### 2.1. Alterations to the Molecular Circadian Clock in the Hypothalamus of Female APP/PS1 Mice

We analyzed the rhythmic expression pattern of clock genes in the 6- and 12-month-old female APP/PS1 mice compared with female control mice (Figure 1). The whole hypothalamus was collected from these groups of mice and qRT-PCR was performed to examine the mRNA expression of *Clock*, *Arntl*, *Cry1*, *Cry2*, *Per1*, *Per2,* and *Per3* at ZT 1, 7, 13, and 19. 

*Clock* mRNA expression exhibited a circadian rhythmicity in 6-month-old wt mice with acrophase in the daytime period, but at 12 months, these control mice lost rhythmicity (Figure 1A,B; Table 1). In APP/PS1 mice, rhythmicity of *Clock* mRNA expression was lost at 6 months, but with aging, transgenic mice showed circadian rhythmicity with acrophase in the dark period (Figure 1A,B; Table 1). *Arntl* mRNA showed a rhythmic pattern in wt and APP/PS1 mice groups at 6-months old with acrophase during light period (Figure 1A,C; Table 1). However, while 12-month-old wt mice lost circadian rhythmicity, it persisted in 12-month-old APP/PS1 mice but with acrophase in the dark phase (Figure 1A,C; Table 1). 

*Cry1* mRNA expression did not show circadian rhythm in wt mice at any age, while 12-month-old APP/PS1 mice showed circadian rhythmicity with acrophase in the dark period (Figure 1A,D; Table 1). *Cry2* mRNA expression only exhibited a circadian rhythmicity in 6-month-old wt mice with acrophase in the daytime period, while this oscillation was observed in 12-month-old APP/PS1 mice with acrophase in the dark period (Figure 1A,E; Table 1). Finally, *Period* genes, *Per1*, *Per2,* and *Per3* showed a similar rhythmic pattern in both mice groups at 6- and 12-months old, with acrophases in the dark period, except *Per1* which exhibited a diurnal acrophase in the 6-month-old mice groups (Figure 1A,F–H; Table 1). Although the oscillatory expression of *Period* genes was kept in 12-month-old APP/PS1 mice, acrophase was delayed in these transgenic mice compared with the control mice (Figure 1A,F–H; Table 1).

Interestingly, gene expression of *Clock* and *Arntl* was significantly higher at ZT 7 in 6-month-old wt mice compared with age-matched APP/PS1 mice (Figure 1B,C). At 12 months, APP/PS1 mice also showed reduced mRNA expression of *Cry1, Cry2, Per1, Per2*, and *Per3* at the nighttime point (ZT 13) compared with age-matched wt mice (Figure 1D–H).

### 2.2. Alterations to the Molecular Circadian Clock in the Hippocampus of Female APP/PS1 Mice

We also examined mRNA expression of clock genes in hippocampal tissue from 6- and 12-month-old APP/PS1 and wt mice, and we found significant alterations in circadian oscillations between both mice groups (Figure 2). 

We found that *Clock* mRNA shows a significant rhythmicity only in 6-month-old wt mice with a diurnal acrophase (Figure 2A,B; Table 2). *Arntl* mRNA showed a rhythmic pattern in 6-month-old wt and APP/PS1 mice, although only the control mice group conserved significant rhythmicity with aging (Figure 2A,C; Table 2). *Cry1* and *Cry2* mRNA exhibited circadian rhythmicity with a diurnal acrophase at 6 months of age and nocturnal acrophase at 12 months of age (Figure 2A,D,E; Table 2). In APP/PS1 mice, *Cry1* and *Cry2* mRNA exhibited circadian rhythmicity with nocturnal acrophase at 6 months of age, which was maintained with aging except in the case of *Cry2* mRNA (Figure 2A,D,E; Table 2). *Per1*-mRNA-only expression showed a rhythmic pattern in both 6-month-old mice groups with an acrophase during the light period; however, acrophase was delayed in the APP/PS1 mice compared with the control mice (Figure 2A,F; Table 2). *Per2* and *Per3* mRNA showed rhythmic pattern in both mice groups at 6- and 12-months old, with nocturnal acrophases except in 6-month-old wt mice who had an acrophase during the light period (Figure 2A,G,H; Table 2).

Contrary to that observed in hypothalamus, gene expression of *Clock* and *Arntl* in the hippocampuses of 6-month-old APP/PS1 mice was increased at ZT 13 compared with age-matched wt mice (Figure 2B,C). In parallel, mRNA expression of *Cry1, Cry2, Per1, Per2*, and *Per3* was also higher in 6-month-old APP/PS1 mice at the nighttime point (ZT 13) compared with age-matched wt mice (Figure 2D–H).

### 2.3. Alterations to the Molecular Circadian Clock in the Cerebral Cortex of Female APP/PS1 Mice

In the cerebral cortex, *Clock* expression is constant except in 6-month-old APP/PS1 mice with acrophase in the daytime period (Figure 3A,B; Table 3). *Arntl* mRNA showed a rhythmic pattern in both 6-month-old mice groups with an acrophase during the light period; although, in APP/PS1 mice it is delayed compared with wt mice (Figure 3A,C; Table 3).

However, the most remarkable effect was the disruption of the circadian rhythmicity of the negative regulators *Cry1, Cry2, Per2*, and *Per3* in 6-month-old APP/PS1 mice. *Cry1, Cry2, Per2*, and *Per3* mRNA expressions showed a rhythmic expression pattern, including acrophase in the dark period, only in 6-month-old wt mice, while this oscillation was absent in age-matched APP/PS1 mice (Figure 3A,D,E,G,H; Table 3). Only *Per1* mRNA displayed a rhythmic pattern in 6-month-old APP/PS1 mice with a diurnal acrophase (Figue 3A,F; Table 3). With aging *Cry1, Cry2, Per2,* and *Per3* oscillation was lost in both mice groups (Figure 3A,D,E,G,H; Table 3).

Similar to that in the hippocampus, *Arntl* expression in the cerebral cortexes of 6-month-old APP/PS1 mice was higher at the daytime point ZT 7 compared with age-matched wt mice (Figure 3C). mRNA expression of *Cry1, Cry2,* and *Per3* was also higher in 12-month-old APP/PS1 mice at the night time point (ZT 19) compared with age-matched wt mice (Figure 2D,E,H).

### 2.4. Aβ Accumulations in the Retinas of Female APP/PS1 Mice

Our group and others have also demonstrated the existence of retinal Aβ accumulation in several AD mouse models [35,36,37]. In contrast, Chidlow et al. did not detect Aβ deposits after immunostaining on retinal sections in APP/PS1 mice at any time point from 3 to 12 months of age [38]. Given this variability from research groups, we deeply investigated Aβ accumulation in retinal samples from 6- and 12-month-old female APP/PS1 and wt mice. As shown in Figure 4A, strong Aβ expression was found in the APP/PS1 retinas of both ages (Figure 4(AII,IV,V)). On the contrary, Aβ staining was not observed in the retina of age-matched wt mice (Figure 4(AI,III)). Intracellular Aβ deposits were detected in retinal ganglion cells (RGCs) but were also detectable in other neuronal classes in the inner nuclear layer (INL) (Figure 4(AIV,V), arrow heads). In the INL, we could observed Aβ immunoreactivity in cell soma in this layer and a long projection to the ganglion cell layer (GCL) (Figure 4(AIV,V), arrow heads). As reported in Figure 4A, immunofluorescent analysis of Aβ staining revealed extensive Aβ accumulation in the soma of RGCs (Figure 4(AVI), arrow heads). Moreover, co-immuno staining of Aβ and lectin revealed retinal vascular Aβ burden in the vessel wall and lumen area, representing Aβ in blood mainly in the GCL but also in the outer nuclear layer (ONL) (Figure 4(AVI,VII), arrows).

Quantitative analysis of Aβ immunoreactivity confirmed significantly increased Aβ deposits in 6- and 12-month-old APP/PS1mice compared with age-matched controls (Figure 4B).

### 2.5. AQPs Expression in the Retinas of APP/PS1 Mice

Because AQP4 facilitates Aβ clearance from the retina [30], we first analyzed retinal AQP4 expression and cellular localization patterns in our mouse model. Previously, retinal expression of *Aqps* was investigated in samples prepared from a large cohort including male and female 6- and 12-month-old wt mice. Since no sex differences were found (Appendix A), both male and female mice were included using mixed-gender groups. The expression of *Aqp8* and *Aqp11* mRNA in both age groups and that of *Aqp0* and *Aqp9* in 12-month-old mice were practically undetectable (Figure 5A), meanwhile low expression of *Aqp7* mRNA was found in retina samples from both mice groups (Figure 5A) However, *Aqp1*, *Aqp3*, *Aqp4*, and *Aqp5* mRNA were clearly expressed in the samples from 6- and 12-month-old wt mice (Figure 5A).

Next, RT-PCR analysis showed that the mRNA levels of *Aqp4* were downregulated in the retinal tissue of APP/PS1 mice at both 6 and 12 months of age (Figure 5B). Since, in the retina, AQP4 is highly expressed in Müller glial cells [39], we also examined *Gs* and *Glast* as Müller glial cell markers. In contrast, the mRNA expression of *Gs* (Figure 5C) and *Glast* (Figure 5D) was not changed by amyloidosis genotype.

By immunofluorescence labeling, AQP4 showed a polarized distribution in retinal Müller cells in wt mice, with great expression in inner retina containing the end foot processes in the GCL, but also in the outer plexiform layer (OPL) (Figure 5E). When the amount of AQP4 staining in the retinal layers was calculated, our findings showed reduced AQP4 expression in the GCL in 6-month-old APP/PS1 mice compared with age-matched wt mice; meanwhile, it was higher in the OPL (Figure 5F).

We found lower levels of *Aqp1* mRNA in the retina of APP/PS1 than of wt mice at 6 and 12 months of age (Figure 6A). In Figure 6B we can see representative images for AQP1 labelled mice retinal sections where AQP1 is expressed at the OPL and the ONL; however, the strong expression was detected in the photoreceptor cell layer (PRL) (Figure 6B). When AQP1 staining was measured in each retinal layer, significantly lower levels of immunoreactivity were found in the APP/PS1 mice compared with the control mice (Figure 6C). In OPL, ONL, and PRL, the retina of 6- and 12-month-old APP/PS1 mice displayed lower levels of AQP1 than those in age-matched control mice, while AQP1 labelling was also lower in the GCL 12-month-old APP/PS1 mice compared with wt mice (Figure 6C). Lower levels of mRNA *Aqp3* were also detected in the retina of the APP/PS1 group than in those the of wt mice at 6 and 12 months of age (Figure 6D). However, when immunohistochemistry analysis was performed on mice retina, no AQP3 immunoreactivity was found. mRNA levels of *Aqp5* were decreased in the retina of the APP/PS1 group compared with those of the wt mice at 6 and 12 months of age (Figure 6E). Additionally, the mRNA expression of *Aqp5* was reduced with aging, with low *Aqp5* expression in both 12-month-old mice groups compared with 6-month-old mice (Figure 6E). Then, we analyzed AQP5 immunoreactivity in mice retina, as shown in Figure 6F. AQP5 expression was detected in practically all retinal cell layers with highly positive signals for GCL, OPL, and PRL (Figure 6F). Interestingly, AQP5 immunoreactivity seemed to colocalize with the cellular retinaldehyde-binding protein (CRALBP), which is expressed both in the RPE and in Müller cells (Figure 6F). Quantitative analysis of AQP5 labelling revealed a significant reduction in the inner (GCL and inner plexiform layer (IPL)) and peripheral (OPL and PRL) layers of retina in 6-month-old APP/PS1 mice compared with wt mice (Figure 6G).

## 3. Discussion

There is particular interest in characterizing the daily rhythmicity of clock genes in female AD mouse models because circadian rhythms are associated with sex differences and female gender is a major risk factor for the development of AD and sex differences in pathological features have been reported in AD [6,7]. Our present results demonstrate a severe disruption of circadian photo-entrainment and pathological changes in the retina of female APP/PS1 mice.

Several important conclusions can be drawn from this study. First, circadian rhythm abnormalities are reported in female APP/PS1 mice. The current study is consistent with our previous research showing the effect of AD on clock gene circadian expression in male transgenic mice [35]. However, the most remarkable finding is the disruption of circadian clock rhythmicity in the hypothalamuses and the hippocampuses of 6-month-old female APP/PS1 mice. The molecular clock is controlled by the expression of the core clock genes *Clock* and *Arntl* as positive regulators; these genes activate the rhythmic transcription of *Per1*–*3* and *Cry1*–*2*, which act as negative regulators [40]. Our present study shows the altered expression of core components of the molecular clock in both the hypothalamus and two extra-hypothalamic brain areas, the hippocampus, and the cerebral cortex, in 6- and 12-month-old female APP/PS1 mice compared to age- and sex-matched control mice. In particular, we were surprised to observe the loss of circadian rhythmicity of *Clock, Cry1,* and *Cry2* in the hypothalamuses of female APP/PS1 mice at 6 months of age. We also observed a reduction in the diurnal gene expression of *Clock* and *Arntl* in these female APP/PS1 mice at 6 months of age. Disruption of critical circadian clock genes not only disrupts the rhythmicity of their expression, but it also has a dramatic effect on the expression levels of their target genes. In young female APP/PS1, hippocampal *Clock* circadian oscillation was also missing, while the gene expression of *Cry1*, *Cry2*, *Per2*, and *Per3* showed an inverted daily pattern. Although in the cerebral cortex rhythmic expression of *Clock* is kept in 6-month-old female APP/PS1 mice, the circadian oscillation of negative regulators was lost. Given these severe disruptions, we suggest that the loss of the rhythmic expression pattern of *Clock* and *Cry1*-*2* in the hypothalamus significantly affects the extra-hypothalamic circadian oscillators. Our findings agree with previous data reported in *Clock*-deficient mice showing that peripheral circadian oscillators require *Clock* [41].

Second, we found that the presence of the neuropathological hallmark Aβ deposits in the retina of these transgenic APP/PS1 mice demonstrates that the retina is specifically affected by neurodegeneration. Aβ deposits have previously been identified in the retina of AD patients [25,26,27,28]. The retina is a CNS tissue that originates in the developing diencephalon and contains a high density of neuronal cells and fibers that form a sensory extension of the brain [42,43]. RGCs project to the brain through the optic nerve and the RHT, the most important pathway for carrying light information to the SCN and for the photoentrainment of circadian rhythm [44]. In APPS/PS1 mice extracellular Aβ deposits have been identified in retinal layers ranging from the NFL to the INL [25,45,46]. Importantly, Aβ accumulation was detected in the retina of APP/PS1 mice as early as 2.5 months of age, 2–3 months prior to their cerebral counterparts [25]. Another study compared plaque burden between male and female mice, and found that in old mice, between 12 and 16 months of age, a significantly greater number of female APPS/PS1 mice exhibited retinal plaque formation compared to age-matched males [45]. Our results show that the most intense Aβ staining occurred in the RGL, as intracellular Aβ accumulation was detected in 6- to 12-month-old APP/PS1 mice. Consistent with our present findings, a recent study by Habiba et al. revealed detectable levels of Aβ_40_ and Aβ_42_ oligomers in the retina and blood as early as 3 months in APP/PS1 mice, long before their detection in the respective part of the brain [47]. Axons of the optic nerve connect the retina to the brain directly and facilitate the transportation of Aβ expressed in RGCs in small transport vesicles [48]. Accordingly with these data, we cannot exclude the possibility that Aβ is initially deposited in the retinal tissue and transported through this optic nerve connection where it will be transported and subsequently accumulated into SCN neurons as we previously reported [35]. We also demonstrated Aβ accumulation in retinal blood vessels in APP/PS1 mice. Similar evidence has been provided in another study showing that Aβ is specifically deposited in the retinal vasculature [49], reinforcing the role of retina as an extension of the brain that is specifically affected by AD. 

In the last decade, several reports have suggested the possibility of a glymphatic system integrated by a vasculature surrounded by perivascular spaces in the eye, proposing that Aβ could be cleared from the retina by perivascular transport [50,51]. More recently, this hypothesis has been supported by evidence of transport from the retina into the CSF [30]. In this last study, the authors supported this fluid transport to facilitate export of waste products from the metabolically active retina. Thus, these two efflux pathways, the ocular and brain glymphatic fluid transport systems, combine, sharing a route of CSF influx along the periarterial space an efflux path along the perivenous space, and a final fluid collection site by the dural and cervical lymphatic vessels [30]. Moreover, these authors demonstrated that Aβ is cleared from the retina via an AQP4-dependent pathway [30]. In the brain glymphatic system, CSF influx and efflux is also facilitated by AQP4 which is located in astrocytic end-feet [52,53,54]. In our present study, we found that *Aqp*4 mRNA levels are downregulated in APP/PS1 retinas, and that the protein was found to be lower specifically in the GCL, suggesting the possibility that the deficiency of AQP4 may compromise the ocular glymphatic system as AQP4 is critical for both glymphatic CSF influx and clearance [55]. A reduction in *Aqp4* mRNA expression has been found in the perivascular zone of the frontal cortex of AD patients [56]. Moreover, deficiency in AQP4 has been shown to impair Aβ clearance through the neurovascular unit, contributing to the progression of AD pathology [57]. As efflux of Aβ is facilitated by AQP4, retinal Aβ accumulation, mainly in the GCL, is consistent with the theory that decreased AQP4-mediated glymphatic clearance may lead retinal neurodegeneration through Aβ accumulation. 

It is widely accepted that AQPs play an important role in maintaining retinal homeostasis in the normal retina. Previous studies have shown that AQP1 is mainly expressed in the PRL and the retinal pigment epithelium (RPE), where it is involved in compensating for altered water transport in this retinal region [58,59]. In addition, a specific type of amacrine cells in the INL may express AQP1 which may be involved in the transmission of visual information from other amacrine or a few cone bipolar cells primarily to ganglion cells [59,60]. Although AQP1 was mainly expressed in the outer retina, it can also be observed in the innermost retinal layers following experimental damage-inducing neural retina edema, and altered glial cell-mediated water transport, diabetes, and ischemia [39,61,62]. In line with these observations, we have shown here that AQP1 expression was significantly decreased in a variety of retinal layers, including GCL, OPL, ONL, and PRL, in APP/PS1 mice. The data suggest that the AQP1-mediated water transport in the retina is altered by this transgenic mouse model of AD. In support of this, AQP1 also facilitates water movement across the blood–CSF barrier as AQP1 is expressed by epithelial cells in the choroid plexus and contributes to the regulation of CSF homeostasis [63,64,65,66,67,68]. A variety of experimental conditions have shown that a reduced AQP1 expression in choroid plexus epithelium can induce lower CSF volume [64,69,70]. Considering all these observations, we suggest that reduced AQP1 expression in the retinal layer of APP/PS1 mice may lead to impaired water transport and signal transduction in the retina.

AQP5 was widely expressed in various retinal layers, from GCL to RPE, suggesting different functions. However, few studies have described the role and distribution of AQP5 in healthy and injured retina beyond its involvement in retinal fluid regulation [71,72]. Interestingly, experimental retinal ischemia results in decreased retinal *Aqp5* gene expression [73]. Downregulation of AQP5 may decrease the water permeability of RPE and thus may inhibit osmotic water flux from the blood into the outer retina; however, it may also impair the retinal fluid clearance [72]. In accordance with these observations, retinal expression of AQP5 was localized in mice and was significantly reduced in APP/PS1 mice. The expression of AQPs, including AQP1, AQP4, and AQP5, has been shown to decrease with age [74,75]. Notably, we also found that *Aqp5* mRNA expression decreased in aged mice compared to young mice, suggesting the involvement of AQP5 in the pathogenesis of age-related retinal diseases, including AD. 

It has been speculated that Aβ may impair retinal homeostasis by altering AQPs expressed by the retinal cells [76]. Aβ induces a highly ordered sequence of molecular and cellular events, involving RGCs, microglial cells, vascular endothelial cells, and photoreceptors. Moreover, the expressions of *Aqp4* mRNA and protein were significantly inhibited in the presence of Aβ, suggesting that Aβ-induced cell damage is due to astrocyte dysfunction through the inhibition of AQP4 [77]. We cannot rule out the possibility that Aβ also affects the expression/distribution of these water channels, and the reduction in retinal AQPs expression we found in our study may be a side effect of retinal Aβ accumulation.

Although the link between the regulation of circadian rhythms and AQPs may not be obvious because it has not been sufficiently studied, some studies have shown this link. A very recent study proposes a relationship between the loss of normal rhythmic expression of circadian proteins, including BMAL1, CLOCK, PER1, PER2, and CRY2, and the polarized expression of AQP4 in perivascular astrocytes [78]. Moreover, authors propose that the circadian gene *Per2* decreases the expression of α-dystrobrevin, a component of the dystrophin-associated complex known to regulate the polarization of AQP4, supporting a link between the perturbation of clock gene expression in the circadian rhythm and AQP4 expression [78]. Furthermore, the connection between circadian regulation and AQPs could be bi-directional as the expression of AQPs may affect the normal circadian rhythm. Indeed, the absence of APQ4 has been shown to exacerbate circadian disruption [79].

The association between circadian regulation and AQPs is not restricted only to the hypothalamic region but has also been described in other peripheral tissues, such as salivary glands, where circadian rhythmic expression of clock genes has been linked to *Aqp5* expression [80]. In addition, *Aqp5* expression in the mouse submandibular gland has been shown to be controlled by the central clock in the hypothalamic SCN [81]. This study reinforces the theory that the expression of AQPs in peripheral organs may be controlled by the central circadian clock in the brain. 

Therefore, in this study, we examined the expression of circadian rhythm clock genes in the brain and the expression of AQPs in the retina, suggesting that both processes may be involved in Aβ accumulation in the retina with neurodegenerative consequences. In conclusion, our findings propose a novel retinal mechanism involved in the clearance of Aβ produced in the RGC mediated by AQP regulation, a mechanism that would be common to both males and females. We speculate that abnormal transport of Aβ, mediated by impaired AQPs expression, contributes to a toxic Aβ accumulation in the retina in the early stages of AD pathology. This allows for impaired transport of retinal signals to the hippocampus, leading to a downregulation of the circadian rhythm mediated by changes in clock gene expression. 

## 4. Material and Methods

### 4.1. Animals

Female and male double-transgenic APP/PS1 mice (6- and 12-month-old), which overexpressed the human genes APP (amyloid beta precursor protein) with the Swedish mutation and exon-9-deleted PSEN1 (presenilin 1; Jackson Laboratory, Bar Harbor; stock no. 005864) were used. Age-matched mice not expressing the transgene were used as wild-type controls (wt). All animals were handled and cared for according to Spanish legislation and guidelines and the Council Directive 2010/63/UE of 22 September 2010, and the ARVO Statement for the Use of Animals in Ophthalmic and Vision Research and ARRIVE guidelines (2020). Mice were maintained in a 12:12 light/dark cycle (Zeitgeber time (ZT)) with *ad libitum* access to food and water and were sacrificed at ZT 1 (08:00), 7 (14:00), 13 (20:00), and 19 (02:00). For molecular and immunohistochemical analysis, animals were sacrificed during light period (~ZT 7) by deep anesthesia and their eyes were immediately dissected and fixed for 24 h in 4% paraformaldehyde (PFA) in 0.1 M phosphate buffer (PB) at pH 7.4.

### 4.2. RNA Extraction and Quantification

At 6 and 12 months of age, mice were euthanized via CO_2_ inhalation at ZT 1, 7, 13, and 19. Brains and eyes were immediately removed and the cerebral cortexes, hippocampuses, hypothalamuses, and retina were extracted and stored at −80 °C. Retinal RNA was obtained by eye dissection from APP/PS1 and wt eys and brain tissues (cerebral cortex, hippocampus, and hypothalamus) using NZYol (NZYTech, Lda., Lisboa, Portugal) following the manufacturer’s protocol. RNA concentration was measured in a NanoDrop™ One Spectrophotometer (Thermo Fisher Scientific, Waltham, MA, USA) and 1µg of each sample was retrotranscribed to cDNA using an iScript™ cDNA Synthesis Kit (Bio-Rad Laboratories, Hercules, CA, USA). Quantitative real-time PCR (qRT-PCR) was performed in a LightCycler^®^ 480 Instrument (Roche Diagnostics, Basel, Switzerland) using NZYSpeedy qPCR Green Master Mix (NZYTech, Lda., Lisbon, Portugal). The primers were predesigned and used in the qRT-PCR to determine the expression levels of *Clock*, *Per*, *Cry*, *Arntl*, *Aqp*, *Gs*, *Glast,* and the housekeeping gene (*Hprt*) (Appendix A). Relative levels of mRNA were calculated using crossing-point (Cp) data and the ΔΔCp method (also known as ΔΔCt). Cp data from the gene of interest (GOI) were normalized to the mean of endogenous gene *Hprt* data to obtain ΔCp data (ΔCp = mean Cp*_Hprt_* − Cp*_GOI_*). ΔΔCp was calculated between the normalized ΔCp values from each time point.

### 4.3. Peroxidase-Based Immunostaining of Aβ

Previously, the fixed eyes were paraffin-embedded, and 4 µm thick sections were collected on microscopy slides using a microtome (Leica, Wetzlar, Germany).

After deparaffinization, treatment with 88% formic acid for 20 min at room temperature was performed in eye sections before peroxidase-based immunostaining for Aβ.

Prior to peroxidase-based immunostaining, the tissues were treated with 3% H_2_O_2_ for 10 min, and the Vectastain Elite ABC HRP kit (Vector Laboratories, Burlingame, CA, USA) protocol was used according to the manufacturer’s instructions. As the primary antibody we used rabbit anti-β-amyloid (ab5076, MerckMillipore, Darmstadt, Germany) which was diluted in phosphate-buffered saline (PBS) 0.1 M containing 5% normal goat serum and 0.5% Triton X-100, and incubated overnight with the tissues at 4 °C. After overnight incubation, primary antibody staining was revealed using one complementary secondary anti-rabbit antibody (HRP conjugated) and 3,3′-diaminobenzidine DAB substrate from the Vectastain Elite ABC HRP kit (SK-4100, Vector Laboratories, Inc., Newark, CA, USA). Counterstaining with Vector hematoxylin (H3401, Vector Laboratories, Burlingame, CA, USA) was performed followed by mounting with DPX (Panreac Quimica, Barcelona, Spain). Images were captured using a light microscope (Zeiss microscope; Carl Zeiss Microimaging, GmbH, Oberkochen, Germany).

### 4.4. Immunofluorescent Staining of Retinal Cross-Sections

For immunofluorescence, one series of 4 µm thick eye sections was used for double-labelling experiments. Previously, and after deparaffinization, eye sections were pre-incubated for 20 min with 88% formic acid at room temperature. Eye sections were then incubated overnight with primary antibodies at 4 °C and diluted in PBS 0.1 M containing 10% normal horse serum and 0.03% Triton X-100. The following primary antibodies were used: rabbit anti-β-amyloid (ab5076, MerckMillipore, Darmstadt, Germany), rabbit-anti AQP1 (ab300463, Abcam, Cambridge, UK), rabbit-anti AQP3 (ab125219, Abcam, Cambridge, UK), rabbit-anti AQP4 (A5971, Sigma-Aldrich, Merck Life Science, Madrid, Spain), and mouse-anti AQP5 (ab78486, Abcam, Cambridge, UK). These antibodies were revealed using fluorescence-conjugated secondary antibodies from Life Technologies: Alexa Donkey anti-mouse 488 (A21202, Molecular Probes, Eugene, OR, USA) and Alexa Goat anti-rabbit 555 (A27039, Molecular Probes, Eugene, OR, USA). DyLight™488 labeled tomato lectin (DL-1174-1, Vector Laboratories, Burlingame, CA, USA), was used as an effective marker of blood vessels. Finally, the slides were mounted with Immunoselect Antifading Mounting Medium with DAPI (SCR-038448, BioTrend, Köln, Germany). Fluorescent images were obtained using a Stellaris Laser Scanning confocal microscope or a Thunder imager wide-field microscope (Leica Microsystems, Wetzlar, Germany) and analyzed using the Image J 1.54f software (U. S. National Institutes of Health, Bethesda, MD, USA). 

Quantitative analysis of the immunostaining was performed using images of the cross-sectional samples of each retinal cell layer until the whole retina had been evaluated. A fluorescence threshold greater than the background was established and the signal above that threshold was measured, obtaining the mean fluorescence parameter.

### 4.5. Statistical Analysis

Data analysis was conducted using GraphPad Prism 6.01 (GraphPad Software, La Jolla, CA, USA) software. All data are expressed as mean ± standard error of the mean (SEM). Multiple comparisons were calculated using two-way ANOVA followed by Bonferroni’s correction. In all cases, statistical significance was set at *p* < 0.05. For the analysis of circadian rhythmicity, we used CircaCompare in RStudio software (version 1.1.419), as previously described [82].

## Figures and Tables

**Figure 1 ijms-24-15679-f001:**
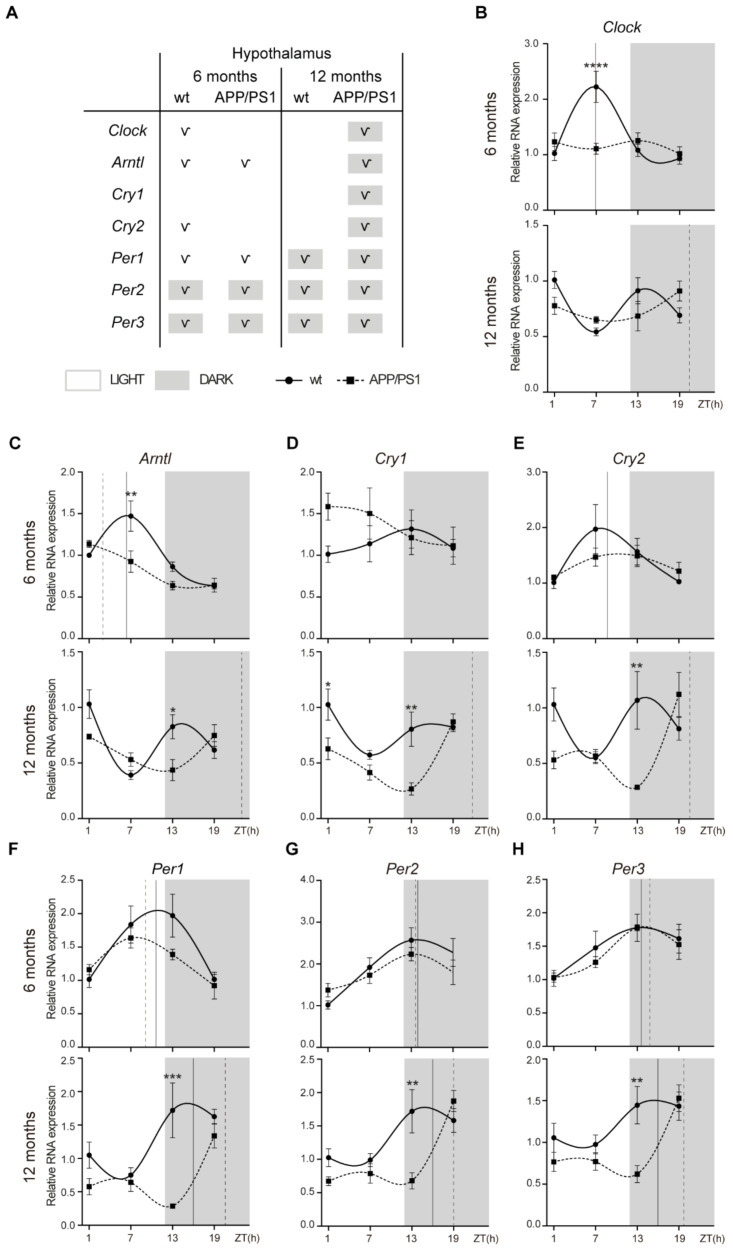
Altered expression profile of clock genes in the hypothalamus of female APP/PS1 mice. (**A**) Schematic table symbolizing the circadian rhythmicity of clock genes, indicating diurnal (light) or nocturnal (dark) acrophase, in the studied mice groups. (**B**–**H**) Transcript levels from wt (filled circles) and APP/PS1 mice (filled squares) 6- and 12-month-old mice at ZT 1, 7, 13, and 19 for *Clock* (**B**), *Arntl* (**C**), *Cry1* (**D**), *Cry2* €, *Per1* (**F**), *Per2* (**G**), and *Per3* (**H**). Dashed lines indicate the time of acrophase for each mice group. We analyzed 4–5 mice per time point and group. Mean ± SEM. * *p* < 0.05, ** *p* < 0.01, *** *p* < 0.001, **** *p* < 0.0001 using two-way ANOVA and Bonferroni’s multiple comparison post-test. wt: wild type, ZT: Zeitgeber Time.

**Figure 2 ijms-24-15679-f002:**
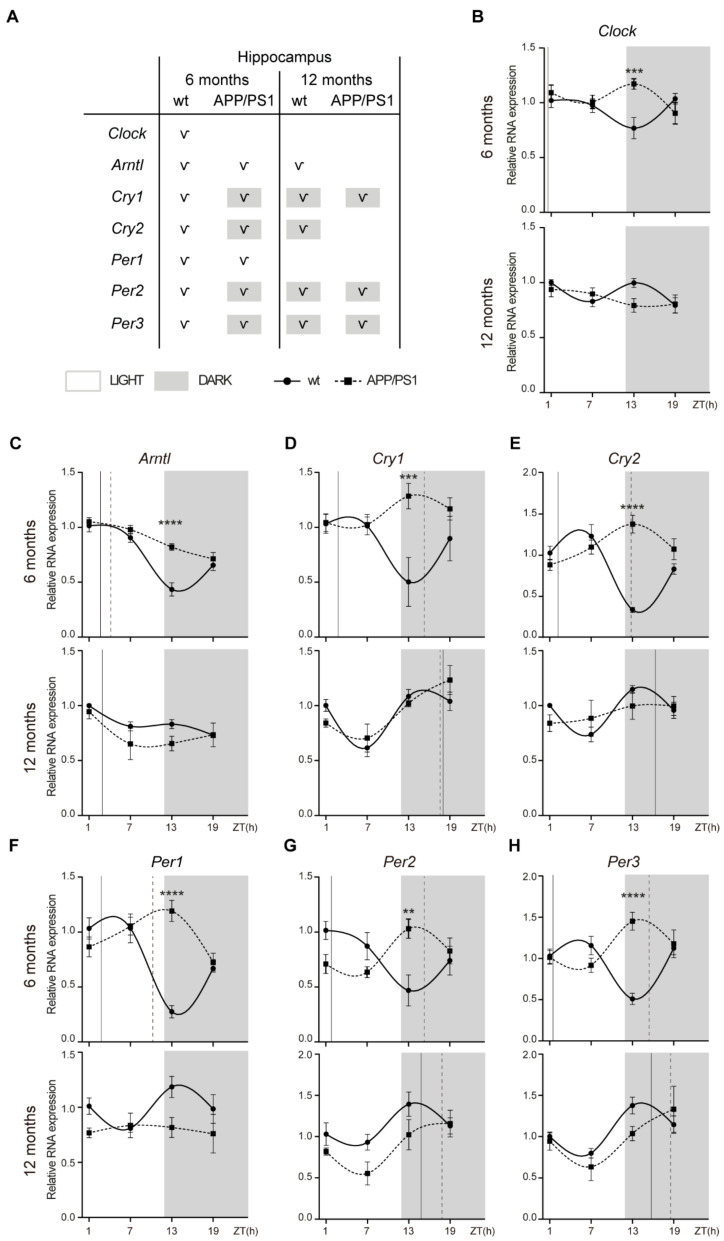
Altered expression profile of clock genes in the hippocampuses of female APP/PS1 mice. (**A**) Schematic table symbolizing the circadian rhythmicity of clock genes, indicating diurnal (light) or nocturnal (dark) acrophase, in the studied mice groups. (**B**–**H**) Transcript levels from wt (filled circles) and APP/PS1 mice (filled squares) 6- and 12-month-old mice at ZT 1, 7, 13, and 19 for *Clock* (**B**), *Arntl* (**C**), *Cry1* (**D**), *Cry2* (**E**), *Per1* (**F**), *Per2* (**G**), and *Per3* (**H**). Dashed lines indicate the time of acrophase for each mice group. We analyzed 4-5 mice per time point and group. Mean ± SEM. ** *p* < 0.01, *** *p* < 0.001, **** *p* < 0.0001 using two-way ANOVA and Bonferroni’s multiple comparison post-test. wt: wild type, ZT: Zeitgeber Time.

**Figure 3 ijms-24-15679-f003:**
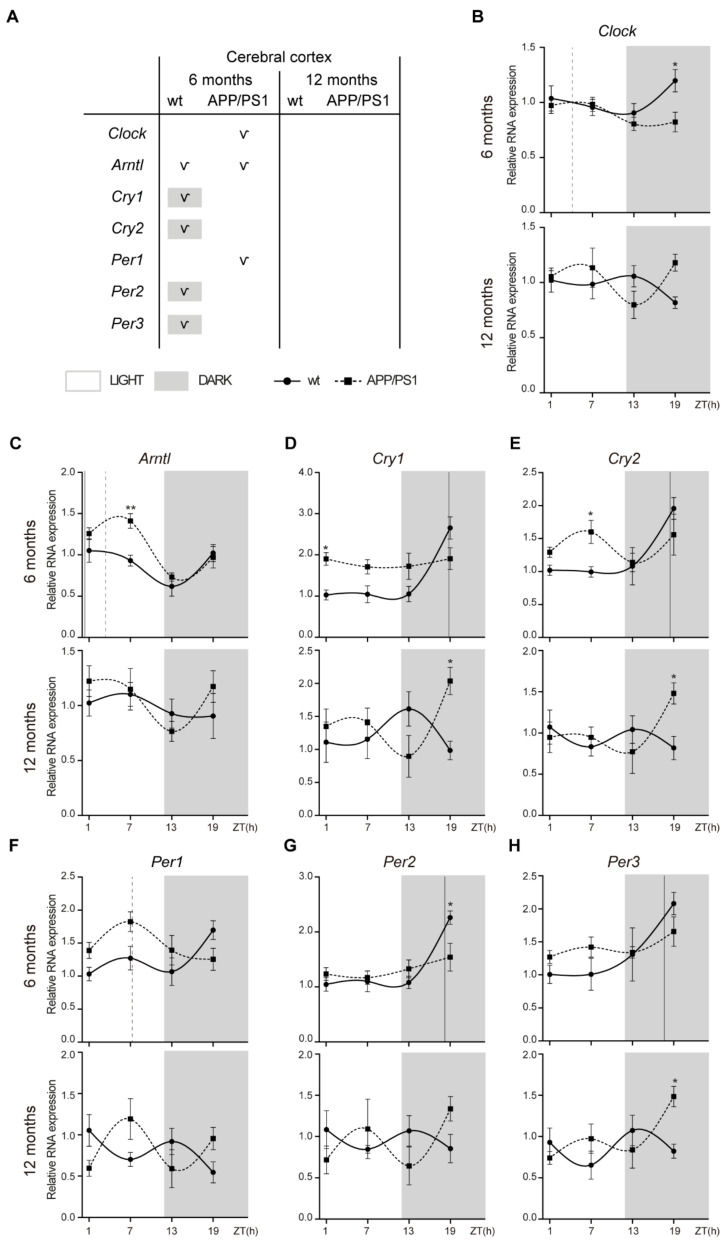
Altered expression profile of clock genes in the cerebral cortex of female APP/PS1 mice. (**A**) Schematic table symbolizing the circadian rhythmicity of clock genes, indicating diurnal (light) or nocturnal (dark) acrophase, in the studied mice groups. (**B**–**H**) Transcript levels from wt (filled circles) and APP/PS1 mice (filled squares) 6- and 12-month-old mice at ZT 1, 7, 13, and 19 for *Clock* (**B**), *Arntl* (**C**), *Cry1* (**D**), *Cry2* (**E**), *Per1* (**F**), *Per2* (**G**), and *Per3* (**H**). Dashed lines indicate the time of acrophase for each mice group. We analyzed 4–5 mice per time point and group. Mean ± SEM. * *p* < 0.05, ** *p* < 0.01 using two-way ANOVA and Bonferroni’s multiple comparison post-test. wt: wild type, ZT: Zeitgeber Time.

**Figure 4 ijms-24-15679-f004:**
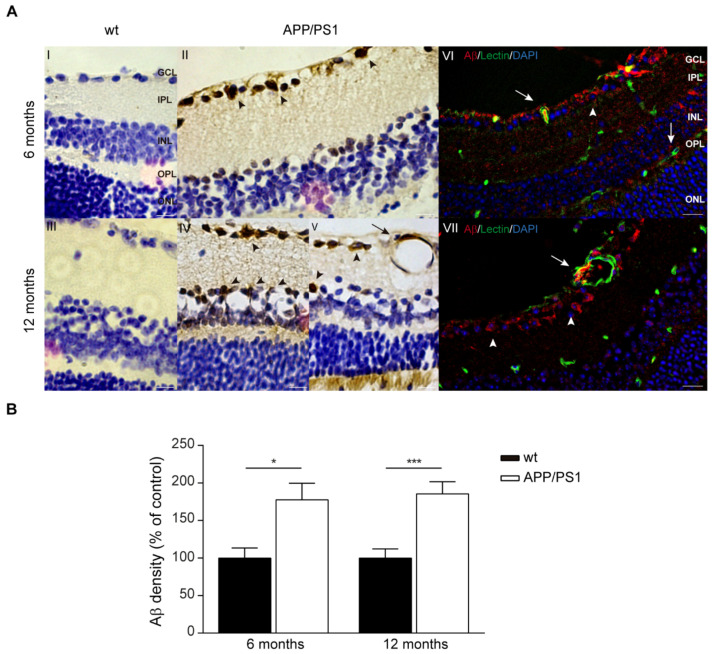
Aβ deposits in retina of female APP/PS1 mice. (**A**) Representative photomicrographs of Aβ deposition in retina of female 6- and 12-month-old APP/PS1 and age-matched wt mice stained with anti-Aβ antibody and visualized with DAB labelling and hematoxylin counterstain (**I**–**V**). In right panels, representative fluorescent images of retina stained for Aβ (red), blood vessels (lectin, green), and nuclei (DAPI, blue) in female 6- and 12-month-old APP/PS1 and age-matched wt mice (**VI**,**VII**). Scale bar = 10 µm. (**B**) Quantitative analysis of Aβ immunostaining showing significant increase in retinal Aβ in female 6- and 12-month-old APP/PS1 and age-matched wt mice. Data expressed as media ± SEM. * *p* < 0.05, *** *p* < 0.001, n = six mice per group, using two-way ANOVA and Bonferroni’s multiple comparison post-test. wt: wild type, GCL: ganglion cell layer, IPL: inner plexiform layer, INL: inner nuclear layer, OPL: outer plexiform layer, ONL: outer nuclear layer.

**Figure 5 ijms-24-15679-f005:**
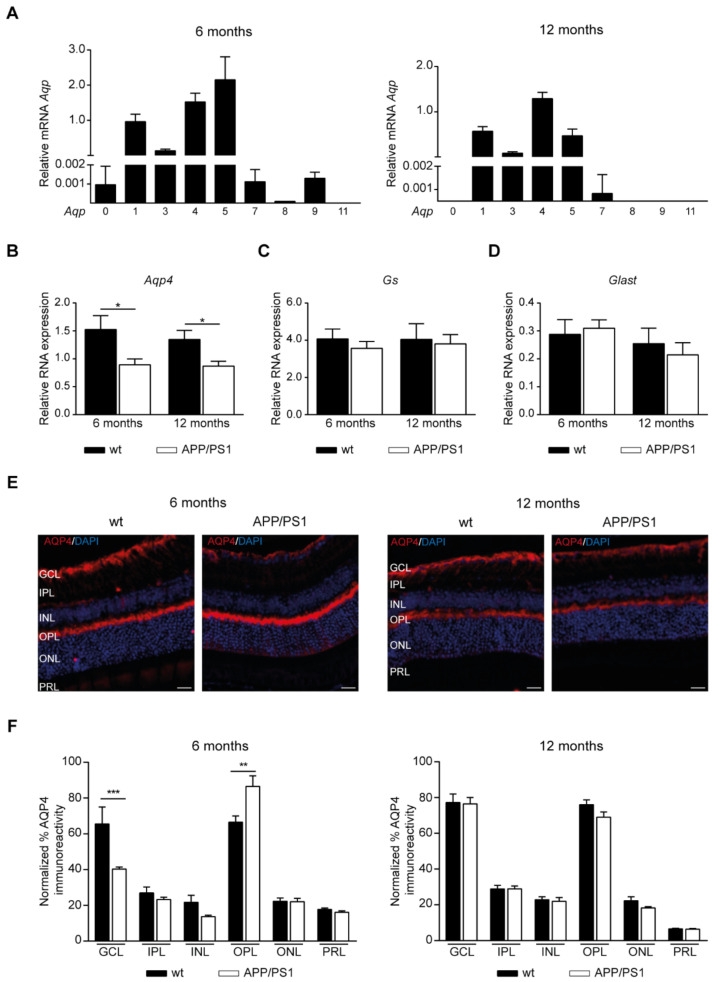
AQP4 localization at the retinal layers of mixed-gender APP/PS1 mice. (**A**) RT-PCR analysis of the differential gene expression of various *Aqps* in retinal samples derived from 6- and 12-month-old wt mice. (**B**–**D**) Alteration in mRNA levels related to *Aqp*4 (**B**), *Gs* (**C**) and *Glast* (**D**) in retinal samples derived from APP/PS1 and wt mice at 6- and 12-months of age. (**E**) Representative retinal sections immunostained with AQP4 in 6- and 12-month-old APP/PS1 and age-matched wt mice. Scale bar = 20 µm. (**F**) Quantitative analysis of AQP4 immunostaining calculated in each retinal layer in each 6- and 12-month-old mice group. Data expressed as media ± SEM. * *p* < 0.05, ** *p* < 0.01, *** *p* < 0.001, *n* = 12 (six male and six female) mice per group, using two-way ANOVA and Bonferroni’s multiple comparison post-test. wt: wild type, GCL: ganglion cell layer, IPL: inner plexiform layer, INL: inner nuclear layer, OPL: outer plexiform layer, ONL: outer nuclear layer, PRL: photoreceptor cell layer.

**Figure 6 ijms-24-15679-f006:**
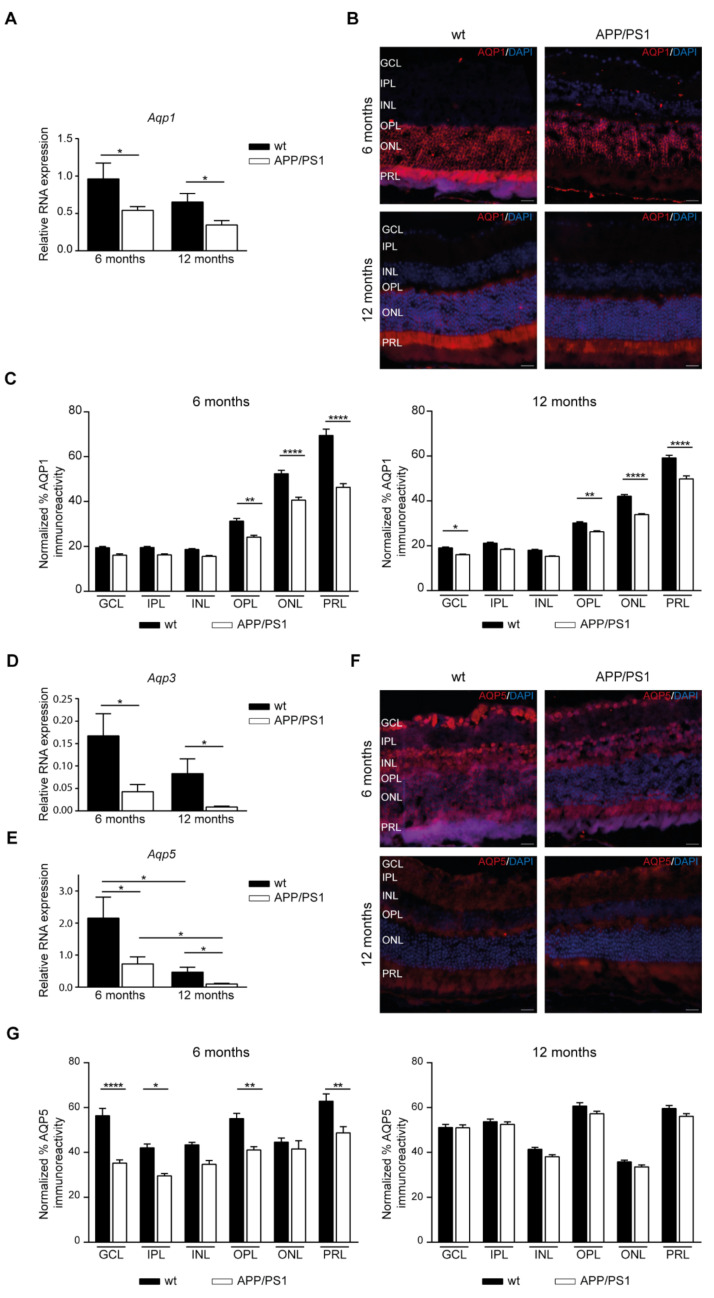
AQPs localization at the retinal layers of mixed-gender APP/PS1 mice. (**A**) Altered mRNA levels of *Aqp1* in retinal samples derived from APP/PS1 and wt mice at 6- and 12-months of age. (**B**) Representative retinal sections immunostained with AQP1 in 6- and 12-month-old APP/PS1 and age-matched wt mice. Scale bar = 20 µm. (**C**) Quantitative analysis of AQP1 immunostaining calculated in each retinal layer in each 6- and 12-month-old mice group. (**D**,**E**) Altered mRNA levels of *Aqp3* (**D**) and *Aqp5* (**E**) in retinal samples derived from APP/PS1 and wt mice at 6- and 12-months of age. (**F**) Representative retinal sections immunostained with AQP5 in 6- and 12-month-old APP/PS1 and age-matched wt mice. Scale bar = 20 µm. (**G**) Quantitative analysis of AQP5 immunostaining calculated in each retinal layer in each 6- and 12-month-old mice group. Data expressed as media ± SEM. * *p* < 0.05, ** *p* < 0.01, **** *p* < 0.0001, n = 12 (six male and six female) mice per group, using two-way ANOVA and Bonferroni’s multiple comparison post-test. wt: wild type, GCL: ganglion cell layer, IPL: inner plexiform layer, INL: inner nuclear layer, OPL: outer plexiform layer, ONL: outer nuclear layer, PRL: photoreceptor cell layer.

**Table 1 ijms-24-15679-t001:** Circadian parameters for clock gene expression in mice hypothalamus.

Gene	Age	Genotype	Period (h)	Phase (h)	Amplitude	*p*-Value
*Clock*	6 months	WT	24.00	6.88	0.68	**5.33 × 10^−4^**
		APP/PS1	24.00	7.69	0.05	0.57
	12 months	WT	24.00	22.64	0.08	0.38
		APP/PS1	24.00	20.38	0.14	**0.04**
*Arntl*	6 months	WT	24.00	6.33	0.43	**5.87 x 10^−5^**
		APP/PS1	24.00	3.05	0.28	**3.94 × 10^−4^**
	12 months	WT	24.00	23.14	0.17	0.15
		APP/PS1	24.00	22.66	0.19	**3.90 × 10^−4^**
*Cry1*	6 months	WT	24.00	12.34	0.16	0.23
		APP/PS1	24.00	4.07	0.27	0.13
	12 months	WT	24.00	21.84	0.16	0.09
		APP/PS1	24.00	21.72	0.30	**2.36 × 10^−4^**
*Cry2*	6 months	WT	24.00	8.81	0.54	**0.02**
		APP/PS1	24.00	10.76	0.23	0.06
	12 months	WT	24.00	18.31	0.10	0.48
		APP/PS1	24.00	20.59	0.30	**0.02**
*Per1*	6 months	WT	24.00	10.34	0.63	**3.78 × 10^−3^**
		APP/PS1	24.00	8.52	0.29	**0.03**
	12 months	WT	24.00	16.50	0.55	**0.02**
		APP/PS1	24.00	20.41	0.40	**0.01**
*Per2*	6 months	WT	24.00	13.88	0.78	**8.88 × 10^−4^**
		APP/PS1	24.00	13.30	0.43	**0.02**
	12 months	WT	24.00	15.69	0.46	**0.01**
		APP/PS1	24.00	18.98	0.54	**1.75 × 10^−4^**
*Per3*	6 months	WT	24.00	13.63	0.37	**0.02**
		APP/PS1	24.00	14.28	0.40	**1.46 × 10^−4^**
	12 months	WT	24.00	16.30	0.30	**0.04**
		APP/PS1	24.00	19.73	0.39	**4.11 × 10^−4^**

wt: wild type; h: hour.

**Table 2 ijms-24-15679-t002:** Circadian parameters for clock gene expression in mice hippocampuses.

Gene	Age	Genotype	Period (h)	Phase (h)	Amplitude	*p*-Value
*Clock*	6 months	WT	24.00	0.35	0.12	**0.02**
		APP/PS1	24.00	9.69	0.04	0.42
	12 months	WT	24.00	8.74	0.02	0.68
		APP/PS1	24.00	3.14	0.09	0.08
*Arntl*	6 months	WT	24.00	2.58	0.32	**2.35 × 10^10^**
		APP/PS1	24.00	4.04	0.17	**2.95 × 10^6^**
	12 months	WT	24.00	2.86	0.09	**0.03**
		APP/PS1	24.00	23.99	0.16	0.05
*Cry1*	6 months	WT	24.00	2.08	0.26	**0.02**
		APP/PS1	24.00	15.26	0.14	**0.047**
	12 months	WT	24.00	18.06	0.22	**2.64 × 10^3^**
		APP/PS1	24.00	17.69	0.28	**9.77 × 10^4^**
*Cry2*	6 months	WT	24.00	3.05	0.39	**2.06 × 10^5^**
		APP/PS1	24.00	12.88	0.24	**7.11 × 10^4^**
	12 months	WT	24.00	16.46	0.14	**0.02**
		APP/PS1	24.00	15.27	0.10	0.24
*Per1*	6 months	WT	24.00	2.79	0.42	**2.20 × 10^6^**
		APP/PS1	24.00	10.17	0.21	**2.56 × 10^3^**
	12 months	WT	24.00	15.76	0.13	0.09
		APP/PS1	24.00	9.25	0.04	0.56
*Per2*	6 months	WT	24.00	1.83	0.29	**5.33 × 10^3^**
		APP/PS1	24.00	15.32	0.20	**2.58 × 10^3^**
	12 months	WT	24.00	14.81	0.22	**0.03**
		APP/PS1	24.00	17.82	0.33	**4.28 × 10^3^**
*Per3*	6 months	WT	24.00	1.40	0.26	**1.76 × 10^3^**
		APP/PS1	24.00	15.38	0.26	**2.16 × 10^3^**
	12 months	WT	24.00	15.71	0.26	**1.72 × 10^3^**
		APP/PS1	24.00	18.51	0.35	**0.01**

wt: wild type; h: hour.

**Table 3 ijms-24-15679-t003:** Circadian parameters for clock gene expression in mice cerebral cortex.

Gene	Age	Genotype	Period (h)	Phase (h)	Amplitude	*p*-Value
*Clock*	6 months	WT	24.00	20.70	0.13	0.07
		APP/PS1	24.00	3.99	0.12	**0.03**
	12 months	WT	24.00	7.64	0.10	0.18
		APP/PS1	24.00	0.23	0.12	0.23
*Arntl*	6 months	WT	24.00	0.26	0.21	**0.02**
		APP/PS1	24.00	3.37	0.31	**3.07 × 10^4^**
	12 months	WT	24.00	5.17	0.11	0.37
		APP/PS1	24.00	0.79	0.22	0.06
*Cry1*	6 months	WT	24.00	18.93	0.74	**7.70 × 10^4^**
		APP/PS1	24.00	21.78	0.13	0.37
	12 months	WT	24.00	11.52	0.29	0.13
		APP/PS1	24.00	21.09	0.37	0.08
*Cry2*	6 months	WT	24.00	18.43	0.45	**1.46 × 10^3^**
		APP/PS1	24.00	4.67	0.07	0.61
	12 months	WT	24.00	5.65	0.04	0.75
		APP/PS1	24.00	19.89	0.29	0.05
*Per1*	6 months	WT	24.00	18.30	0.18	0.22
		APP/PS1	24.00	7.22	0.30	**0.01**
	12 months	WT	24.00	5.52	0.14	0.31
		APP/PS1	24.00	7.81	0.12	0.42
*Per2*	6 months	WT	24.00	18.23	0.54	**1.01 × 10^3^**
		APP/PS1	24.00	17.89	0.18	0.10
	12 months	WT	24.00	5.98	0.03	0.85
		APP/PS1	24.00	19.16	0.12	0.52
*Per3*	6 months	WT	24.00	17.67	0.53	**9.38 × 10^6^**
		APP/PS1	24.00	14.60	0.09	0.45
	12 months	WT	24.00	15.17	0.09	0.47
		APP/PS1	24.00	17.82	0.24	0.09

wt: wild type; h: hour.

## Data Availability

Data are contained within the article.

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
