# Peer review of "Altered Clock Gene Expression in Female APP/PS1 Mice and Aquaporin-Dependent Amyloid Accumulation in the Retina"

_ijms, 2023, doi:10.3390/ijms242115679_

Round 1
Reviewer 1 Report
I don´t find the paper suitable for special topic "Aquaporins in Brain Disease" in it´s current form. The paper is mostly about circadian regulation (shown only by gene expression profiles) and much less about aquaporins. The connection of circadian regulation and aquaporins is not apparent. It is also strange to emphasize the female sex so much in the main title and introduction, but then not compare the sexes in the data! In a matter of fact, the authors have already published a similar work before, but only in male mice. In that paper, the authors do not emphasize male sex. The are several flaws in the study design and all the conclusions are not justified from the data. For example, authors state "retinal neurodegeneration" in the abstract, but it is not evaluated in the manuscript. In fact, the retinal layers look by naked eye identical between WT and AD mice to me (Figures 5-6).
I think it would be better to detach the circadian regulation story from the aquaporin characterization story, and do the aquaporin characterization in a much more detailed fashion, for example associating it with the disease progression and comparing females and males.
English was fine and text was intelligible. However, it is not 100 % standand English. Proof-reading could increase quality.
Author Response
Reviewer 1
I don´t find the paper suitable for special topic "Aquaporins in Brain Disease" in it´s current form. The paper is mostly about circadian regulation (shown only by gene expression profiles) and much less about aquaporins. The connection of circadian regulation and aquaporins is not apparent. It is also strange to emphasize the female sex so much in the main title and introduction, but then not compare the sexes in the data! In a matter of fact, the authors have already published a similar work before, but only in male mice. In that paper, the authors do not emphasize male sex. The are several flaws in the study design and all the conclusions are not justified from the data. For example, authors state "retinal neurodegeneration" in the abstract, but it is not evaluated in the manuscript. In fact, the retinal layers look by naked eye identical between WT and AD mice to me (Figures 5-6).
I think it would be better to detach the circadian regulation story from the aquaporin characterization story and do the aquaporin characterization in a much more detailed fashion, for example associating it with the disease progression and comparing females and males.
We appreciate the reviewer comment. Hence new paragraphs explaining connection of circadian regulation and aquaporins have been included in the Discussion section. (pages 29-30) in the revised version of manuscript.
Briefly, a very recent study propose a relation between the loss of the normal rhythmic expressions of circadian proteins Per2, Cry2, Bmal1, Clock, and Per1, and the polarized expression of aquaporin-4 (AQP4) in perivascular astrocytes (Yao et al, Transl Psychiatry. 2023; 13: 310). AQP4 facilitates the functioning of the glymphatic system, involved in clearing soluble materials, including Aβ. Moreover, in this study (Yao et al, 2023), authors propose that the circadian gene Per2 diminishes the expression of Dtna, a component of the DAC known to regulate the polarization of AQP4, supporting a link between the perturbation of clock gene expression in the circadian rhythm and AQP4 expression.
Furthermore, the connection between circadian regulation and aquaporins could be bi-directional as expression of AQPs may affect normal circadian rhythm. Indeed, it has been shown that AQP4 absence exacerbates the prolonged light-induced impairment of circadian oscillations and delays their recovery to normal rhythmicity (Murakami et al, J Biol Rhythms. 2023 Apr;38(2):208-214).
This association between circadian regulation and AQPs is not restricted only to hypothalamic region but has also been described in other peripheral tissue, such as salivary glands, where circadian rhythmic expression of clock genes has been associated with AQP5 expression (Satou et al, Front Physiol. 2017 May 23:8:320). Additionally, it has been shown that Aqp5 expression in the mouse submandibular gland is driven by the central clock in the hypothalamic suprachiasmatic nucleus (Uchida et al, J Physiol Sci. 2018 Jul;68(4):377-385). This study reinforces the theory that AQPs expression in peripheral issues may be controlled by the central circadian clock in the brain.
Thus, in this study we evaluated the expression of circadian rhythm clock genes in the brain and the expression of AQPs in retina proposing that both processes could be involved in Aβ accumulation in retina driven neurodegenerative consequences.
We also agree and appreciate the reviewer comments including some “non-well written” sentences” as “retinal neurodegeneration” in the abstract. We have changed this sentence in the revised version of manuscript trying to be more precise in the aim of our present work. In this study, we proposed to focus on the key pathological hallmark of AD, Aβ deposits, in retina as mediator of retinal degeneration, a mechanistic signaling supported by many findings (Hart N.J., et al, Acta Neuropathol. 2016;132:767–787; La Morgia et al, Ann Neurol 2016, 79, 90-109; Koronyo, Y et al, JCI Insight 2017, 2; den Haan, J. et al, Acta Neuropathol Commun 2018, 6, 147). Accordingly with these studies, our results showed Aβ deposits in retinal layers, including GCL and INL, in 6- and 12-month-old female APP/PS1 mice, whereas it was absent in wt mice. In parallel, we also found statistically significant differences in AQPs expression, including AQP1, AQP3, AQP4 and AQP5, in retina in 6- and 12-month-old female APP/PS1 mice compared to age-matched control mice.
English was fine and text was intelligible. However, it is not 100 % standand English. Proof-reading could increase quality.
We thank the reviewer suggestion and English revision has been made.
Reviewer 2 Report
The article titled:” Altered clock gene expression in female APP/PS1 mice and Aquaporin-dependent amyloid accumulation in the retina” by Laura Carrero et al. studies circadian rhythm disturbances and retinal neurodegeneration in Alzheimer´s Disease (AD) considering the gender female as a main risk factor.
The article reviews the effects of AD progress on genes involved in circadian rhythm control using an animal model and focusing its studies on females. The data are conclusive since there is a specific alteration in the expression pattern of these genes. They also evaluate the alterations detected in the retina, and fundamentally analyze the expression of AQP proteins. The authors define APQ4 and 5 as involved in the degenerative processes associated with AD detected in the retina.
It is an interesting article, novel with interest for the scientific community and with possible repercussions in future studies.
Some comments
-Please check the phrases on line 50 and 52, with similar and could be unified.
"Circadian rhythm disturbances are associated with the functional disruption of biological clocks in the suprachiasmatic nucleus (SCN) " and "Circadian rhythms are orchestrated by a brain clock located in the hypothalamus SCN".
-In the results section, line 92: please indicate whether the controls are male or female.
-In the results section, line 182: please check the beginning of the paragraph as something seems to be missing.
-In the results section, line 192: the authors comment on the detection of A deposits in Müller glia cells, as well as in horizontal cells or amacrins. This type of association may be undefined and not corroborated. You could include the location by referring to the layers of the retina, but it is not shown which type is involved.
- Please indicate the group of mice and their corresponding ZT, which have been used to make molecular and immunohistological approaches in the eye.
-Figure 5: It would be interesting to note that the retinas of the group of APP/PS1 mice present histological alterations in both cytoarchitecture and thickness of the retinal layers. These changes are consistent with AD-associated retinal neurodegeneration. On the other hand, in the text (line 232), the authors comment that the activation of Müller’s glia does not correlate with a higher level of mRNA. This fact is within logic, because normally the number of cells is not increased, they are the same, only that active (reactive gliosis) with projections, and detected in layers of the retina mainly damaged.
In panel F, in figure 5, it is recommended to improve the selected images, as they are not sharp, and the fact of including the autofluorescence in green, or underlying marking, can make it difficult to interpret the information about AQP4.
-Figure 6: in panel F, it is recommended to discuss the fact that the marking is compatible with the location of the CRALBP, as an astroglia marker, and to reference the colocalization of AQP5.
The images shown in panel B and F differ in size, either the scale is not the same, or they are images taken in different areas of the retina (peripheral in B and central in F). It is suggested to show images that are comparable.
-Material section and methods: Please, line 402, the authors have put in parentheses (retina), but have actually enucleated the eyes, and made sections of them. Retina, it would be if they dissected the retina from the eye ball. Line 433, please indicate the thickness of the eye sections.
Author Response
Reviewer 2
The article titled:” Altered clock gene expression in female APP/PS1 mice and Aquaporin-dependent amyloid accumulation in the retina” by Laura Carrero et al. studies circadian rhythm disturbances and retinal neurodegeneration in Alzheimer´s Disease (AD) considering the gender female as a main risk factor.
The article reviews the effects of AD progress on genes involved in circadian rhythm control using an animal model and focusing its studies on females. The data are conclusive since there is a specific alteration in the expression pattern of these genes. They also evaluate the alterations detected in the retina, and fundamentally analyze the expression of AQP proteins. The authors define APQ4 and 5 as involved in the degenerative processes associated with AD detected in the retina.
It is an interesting article, novel with interest for the scientific community and with possible repercussions in future studies.
Some comments
-Please check the phrases on line 50 and 52, with similar and could be unified.
"Circadian rhythm disturbances are associated with the functional disruption of biological clocks in the suprachiasmatic nucleus (SCN) " and "Circadian rhythms are orchestrated by a brain clock located in the hypothalamus SCN".
We thank reviewer indication, and the second (redundant) phase has been eliminated in the revised version of manuscript.
-In the results section, line 92: please indicate whether the controls are male or female.
Following the reviewer indication, “female control mice” has been indicated in the first paragraph of Results Section in the revised version of manuscript.
-In the results section, line 182: please check the beginning of the paragraph as something seems to be missing.
We apologize the mistake, and the missing word (“Our”) has been included in the revised version of manuscript.
-In the results section, line 192: the authors comment on the detection of A deposits in Müller glia cells, as well as in horizontal cells or amacrins. This type of association may be undefined and not corroborated. You could include the location by referring to the layers of the retina, but it is not shown which type is involved.
We thank reviewer suggestion, and the sentence has been rephased in the revised version of manuscript.
- Please indicate the group of mice and their corresponding ZT, which have been used to make molecular and immunohistological approaches in the eye.
As indicated by the reviewer, information about mice used in molecular and immunohistological approaches in the eye has been included in the revised version of manuscript (Material and Methods section, page 31).
-Figure 5: It would be interesting to note that the retinas of the group of APP/PS1 mice present histological alterations in both cytoarchitecture and thickness of the retinal layers. These changes are consistent with AD-associated retinal neurodegeneration. On the other hand, in the text (line 232), the authors comment that the activation of Müller’s glia does not correlate with a higher level of mRNA. This fact is within logic, because normally the number of cells is not increased, they are the same, only that active (reactive gliosis) with projections, and detected in layers of the retina mainly damaged.
In panel F, in figure 5, it is recommended to improve the selected images, as they are not sharp, and the fact of including the autofluorescence in green, or underlying marking, can make it difficult to interpret the information about AQP4.
As indicated by the reviewer, this figure 5 has been changed in the revised version of manuscript.
-Figure 6: in panel F, it is recommended to discuss the fact that the marking is compatible with the location of the CRALBP, as an astroglia marker, and to reference the colocalization of AQP5.
The images shown in panel B and F differ in size, either the scale is not the same, or they are images taken in different areas of the retina (peripheral in B and central in F). It is suggested to show images that are comparable.
As recommended the reviewer, potential colocalization of AQP5 with CRALBP in the figure 6 has been indicated in the revised version of manuscript.
As indicated by the reviewer, this figure 6 has been changed in the revised version of manuscript.
-Material section and methods: Please, line 402, the authors have put in parentheses (retina), but have actually enucleated the eyes, and made sections of them. ???? Retina, it would be if they dissected the retina from the eye ball. Line 433, please indicate the thickness of the eye sections.
I apologize for the confusion. For RNA extraction retina was extracted but it was not sectioned. Additional retina samples were used for immunohistochemistry approach and for that, retina sections were made.
Following the reviewer indication, the thickness of the eye sections has been indicated in the line 402 of the revised version of manuscript.
Round 2
Reviewer 1 Report
In the future work, I suggest to include both sexes in the same manuscript, and compare them directly. Would increase scientific quality and mitigate confusion by the readers.